# Spatial clustering of notified tuberculosis in Ethiopia: A nationwide study

**Kefyalew Addis Alene**[1,2,3]*, **Archie C. A. Clements**[1,2]

**1** Faculty of Health Sciences, Curtin University, Perth, Western Australia, Australia, **2** Wesfarmers Centre of Vaccines and Infectious Diseases, Telethon Kids Institute, Perth, Western Australia, Australia, **3** Institute of Public Health, College of Medicine and Health Sciences, University of Gondar, Gondar, Ethiopia

* kefyalew.alene@telethonkids.org.au

**Data Availability Statement:** All relevant data are within the paper and its Supporting Information files.

**Funding:** The authors received no specific funding for this work.

## Abstract

### Background

Tuberculosis (TB) remains a major health problem worldwide and in Ethiopia. This study aimed to investigate the spatial distributions of notified TB over the whole territory of Ethiopia and to quantify the role of health care access, environmental, socio-demographic, and behavioural factors associated with the clustering of TB.

### Methods

A spatial analysis was conducted using national TB data reported between June 2016 and June 2017 in Ethiopia. Spatial clustering of TB was explored using Moran's I statistic and the local indicator of spatial autocorrelation (LISA). A multivariate Poisson regression model was developed with a conditional autoregressive (CAR) prior structure and with posterior parameters estimated using Bayesian Markov chain Monte Carlo (MCMC) simulation with Gibbs sampling to investigate the drivers of the clustering.

### Result

A total of 120,149 TB cases were reported from 745 districts in Ethiopia during the study period; 41,343 (34%) were bacteriologically confirmed new pulmonary TB and 33,997 (28%) were clinically diagnosed, new, smear-negative pulmonary TB patients. The nationwide annual incidence rate of notified TB was 112 per 100,000 population. The highest incidence was observed in three city administrative regions, namely Dire Dewa (348 cases per 100,000 population), Addis Ababa (262 per 100,000 population) and Harari (206 per 100,000 population), and the lowest incidence was observed in Somali region (51 per 100,000 population). High-high spatial clustering of notified TB was detected at Humera, Gog, and Surima district, and low-low clustering was detected in some districts located in the Somali region. Poor health care access (IRR = 0.78; 95%CI: 0.66, 0.90) and good knowledge about TB (IRR = 0.84; 95%CI: 0.73, 0.96) were negatively associated with the incidence of notified TB.

**Competing interests:** The authors have declared that no competing interests exist.

## Conclusion

Substantial spatial clustering of notified TB was detected at region, zone and district level in Ethiopia. Health care access and knowledge about TB was associated with incidence of TB. This study may provide policy makers target hotspot areas, where national control programs could be implemented more efficiently for the prevention and control of TB, and to address potential under-reporting in poor access areas.

## Introduction

Despite being preventable and treatable, tuberculosis (TB) remains the leading cause of death from an infectious disease worldwide, killing more than one million people every year. According to the WHO's 2018 report, there were an estimated 10 million cases of TB and 1.6 million deaths due to TB globally in 2017 [1]. The magnitude of the disease varies by country, with low and middle-income countries most affected [2]. Approximately 87% of the world's TB cases occur in the 30 high TB burden countries [3]. The African continent accounts for one quarter (25%) of the world's TB cases and the highest proportion of TB cases co-infected with human immunodeficiency virus (HIV) [1].

Ethiopia is one of the 30 designated high TB burden countries globally, with an estimated 170,000 cases and 25,000 deaths due to TB in 2017 [3]. The country has been implementing the directly observed treatment, short course (DOTS) strategy since 1994 [4, 5], and recently adopted the global end-TB strategies. Clear information about locations with the greatest TB burden is needed to guide tailored interventions and target preventive and treatment measures.

Previous studies conducted in Ethiopia reported spatial clustering of TB at a sub-national level and identified ecological factors associate with TB clustering [6, 7]. However, the studies were limited only to some parts of the country and did not assess the effects of health care access, knowledge about TB, or behavioural factors on clustering of TB. To date, no nation-wide spatial analysis of TB incidence has been reported in Ethiopia.

The aim of this study was to quantify the spatial patterns of notified TB in Ethiopia at district level, using a Bayesian statistical framework, and to quantify the role of environmental, socio-demographic, behavioural, and health care access factors in influencing the distribution of the disease.

## Methods

### Study area

The study was conducted in Ethiopia, the second most populous country in Africa, with a population size of around 100 million [8]. The country is administratively divided into 9 regions and 2 administrative cities (Addis Ababa and Dire Diwa), and has a tiered administrative system consisting of regional states (first-level), zones (second-level), districts (woreda), and neighbourhoods (kebeles). The districts are the third-level administrative divisions where TB data are aggregated and reported to the national level.

### Study design and data sources

This ecological study used nationwide notified TB data, obtained from the Health Management Information System (HMIS) in the Ethiopian Ministry of Health. The study included all

forms of TB (including bacteriologically confirmed new pulmonary TB, clinically diagnosed new smear negative pulmonary TB, clinically diagnosed extra pulmonary TB, bacteriologically confirmed relapse pulmonary TB, TB after relapse, TB after treatment failure and TB after loss to follow up) that had been reported to the HMIS from all district of Ethiopia, during the period June 2016 to June 2017. Population estimates used in this study were obtained from the Ethiopia Central Statistical Agency (CSA) projection report [8]. A wide range of socioeconomic, environmental, behavioural, and health care access data were collected from different sources, including from the Ethiopia Demographic and Health Survey (EDHS), EDHS spatial data repository, and the Atlas of Population Density (S1 Table). Data on the knowledge and attitudes of the population regarding TB were obtained from the 2011 EDHS. Data on health care access and behavioural factors such as chat chewing, and alcohol drinking were obtained from the 2016 EDHS. Socio-economic data such as population density, dependency ratio, average number of people in the household, unemployed rate, and literacy rate were obtained from the Ethiopia Atlas of Population Density, and the wealth index was obtained from the 2016 EDHS. Climatic and environmental data such as Enhanced Vegetation Index, rainfall, aridity, and mean temperature were obtained from the EDHS Spatial Analysis repository.

## Data analysis

Statistical analyses were performed in stages. First, the standardized morbidity ratio (SMR) of notified TB for each district was calculated by dividing the observed number of TB cases by the expected number, which was calculated as the product of overall incidence of TB and the average population for each district during the study period. We grouped all forms of TB into a single outcome. A separate analysis was also performed for bacteriologically confirmed new TB only because smear-positive TB cases are the largest source of new TB infections in the community, and this analysis could provide clues about the transmission of TB in the districts.

Second, exploratory spatial analysis was conducted with the Global Moran's I test and the local indicator of spatial autocorrelation (LISA) to assess for the presence of spatial autocorrelation and for the detection of hotspot areas, respectively.

Third, univariable Poisson regression models were built to select covariates appropriate for the spatial analysis. All significant variables in the univariable analysis, at an alpha level = 0.2, were tested for multicollinearity and those variables with a variance inflation factor (VIF) less than 6 were selected for the final spatial model.

Fourth, a multivariate Poisson regression model was developed with the covariates selected during the univariable modelling stage, a spatially unstructured random effect, and a spatially structured random effect designed using a conditional autoregressive (CAR) prior structure. [9] Posterior parameter distributions were estimated using a Bayesian Markov chain Monte Carlo (MCMC) simulation approach with Gibbs sampling, in the WinBUGS statistical software (WinBUGS, MRC Statistical Unit, Cambridge, UK). The complete model was constructed as follows:

$$Y_i \sim \text{Poissson}(\mu_i);$$

$$Log\ (\mu i) = log\ (Ei) + \alpha + \beta k\ Xik + Ui + Vi;$$

where $Y_i$, the observed number of notified TB cases in district $i$, was assumed to follow a Poisson distribution with mean $\mu_i$; $E_i$ was the expected number of notified TB cases in district $i$, $\alpha$ was the intercept; $\beta_k$ was the coefficient for covariate $X_k$, $Ui$ were unstructured random effects and $V_i$ were spatially structured random effects. To define the CAR prior structure of the spatially structured random effects, a neighbourhood weights matrix was developed using the

queen definition, whereby two areas were considered as neighbours if they shared a common boundary or vertex. Weights were allocated a value of 1 if a pair of districts were neighbouring and otherwise a value of zero. Non-informative priors were specified for the intercept $\alpha$ (a non-informative, improper prior with bounds $-\infty$ and $+\infty$) and the coefficients (normal prior with mean = 0 and precision $1 \times 10^{-6}$). The priors for the precision of the unstructured and spatially structured random effects were assigned non-informative gamma distributions with shape and scale parameters set at 0.001. Multiple variations of the model were fitted, including versions without the structured and unstructured effects, and covariates, to establish whether inclusion of these components improved model fit. The models were run for 1,000,000 iterations and convergence was successfully achieved after 900,000 iterations for each of the models. Convergence of the models was determined by visual inspection of posterior kernel density and history plots. The best fitting model was selected based on the deviance information criteria (DIC), whereby a model with a lower DIC value was selected and reported as a better-fitting, more parsimonious model.

### Ethics clearance

This study was approved by the Australian National University Human Research Ethics Committee (protocol number 2016/218) and a letter of endorsement was obtained from the Ethiopian Ministry of Health.

## Results

Table 1 shows the summary statistics of notified TB in Ethiopia reported from 745 districts for the period June 2016 to June 2017. A total of 120,149 TB cases were reported during the study period; 41,343 (34%) were bacteriologically confirmed new pulmonary TB and 33,997(28%) were clinically diagnosed, new, smear-negative pulmonary TB patients. The national annual incidence rate of notified TB was 112 per 100,000 population. The highest incidence rate of notified TB was observed in three city administration regions, namely Dire Dewa (348 cases per 100,000 population), Addis Ababa (262 per 100,000 population) and Harari (206 per 100,000 population); whereas the lowest notified TB incidence rate was observed in Somali region (51 per 100,000 population) (Fig 1).

The incidence rate of bacteriologically confirmed and all forms of notified TB at zone level are presented in the appendix S2 Table. We observed spatial variation in the notified TB incidence rate at the district levels. Fig 2 shows the distribution of standardized morbidity ratios

**Table 1. Number and percentage of tuberculosis patients reported in Ethiopia stratified by type of tuberculosis.**

| Type of TB | Number | Percent |
|---|---|---|
| Bacteriologically confirmed new pulmonary TB | 41343 | 34.4 |
| Clinically diagnosed new smear negative pulmonary | 33997 | 28.3 |
| Clinically diagnosed new extra pulmonary TB | 37557 | 31.3 |
| Bacteriologically confirmed relapse pulmonary TB | 2557 | 2.1 |
| Tuberculosis treatment after relapse | 3162 | 2.6 |
| Tuberculosis treatment after failure | 421 | 0.4 |
| Tuberculosis treatment after loss to follow-up | 325 | 0.3 |
| Tuberculosis other* | 627 | 0.5 |
| MDR-TB | 160 | 0.1 |
| Total | 120149 | |

* Previously treated unknown and undocumented treatment outcome

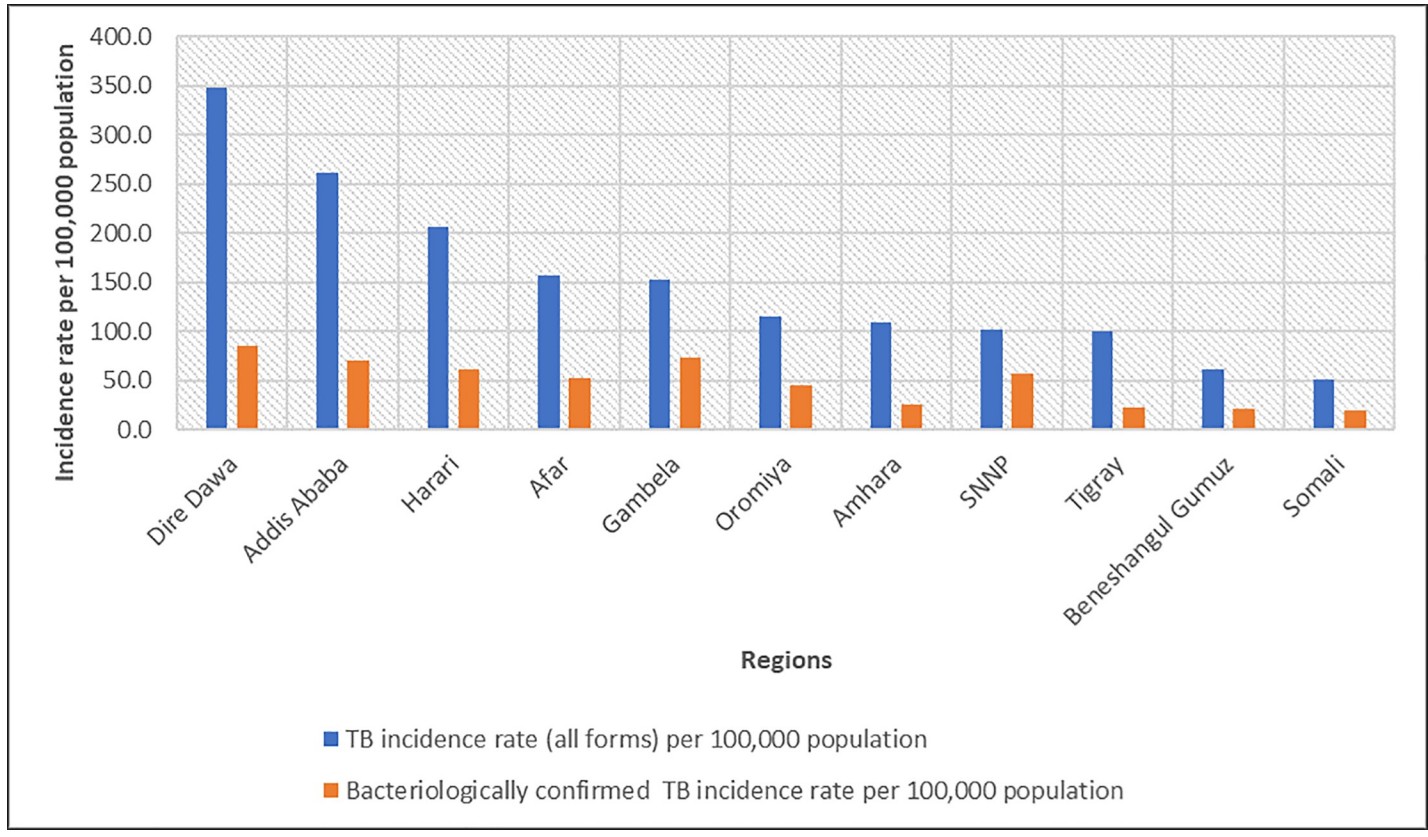

**Fig 1. Tuberculosis notification rate per 100,000 population by regions in Ethiopia, 2016–2017.**

(SMR) of TB at district level in Ethiopia. The highest SMR of TB were found in districts located in the central part of the country, and in districts near to the regional cities. The SMR of notified TB were relatively low in the eastern part of the country, in Somali Region.

## Spatial clustering of notified TB

The initial exploratory analysis showed positive global spatial autocorrelation (with a Moran's I index = 0.144; p-value = 0.017) and local spatial clusters of TB (Fig 3). In the LISA analysis, districts located in Somali Region showed a low-low type of relationship, meaning that these districts had a low incidence of notified TB and the surrounding districts had also low notified TB incidence (Fig 3). A high-high cluster of notified TB was found in northwest Ethiopia, which is located on the Ethiopia and Eritrea border. The maps of bacteriologically confirmed new pulmonary TB showed a low-low clustering of notified TB in Amhara, Tigray, and Somali regions. A high-high cluster of bacteriologically confirmed TB was also found in districts located in Gambela and Southern regions (Fig 3).

## Factors associated with spatial clustering of notified TB

Table 2 shows the Bayesian multivariable Poisson regression model of ecological-level factors associated with notified incidence of TB in Ethiopia. The model with spatially structured random effect (excluding the unstructured random effect) was the best-fitting of the Bayesian models. It showed that the incidence of notified TB was significantly associations with poor health care access (IRR = 0.78; 95%CI: 0.66, 0.90), and good knowledge about TB (IRR = 0.84;

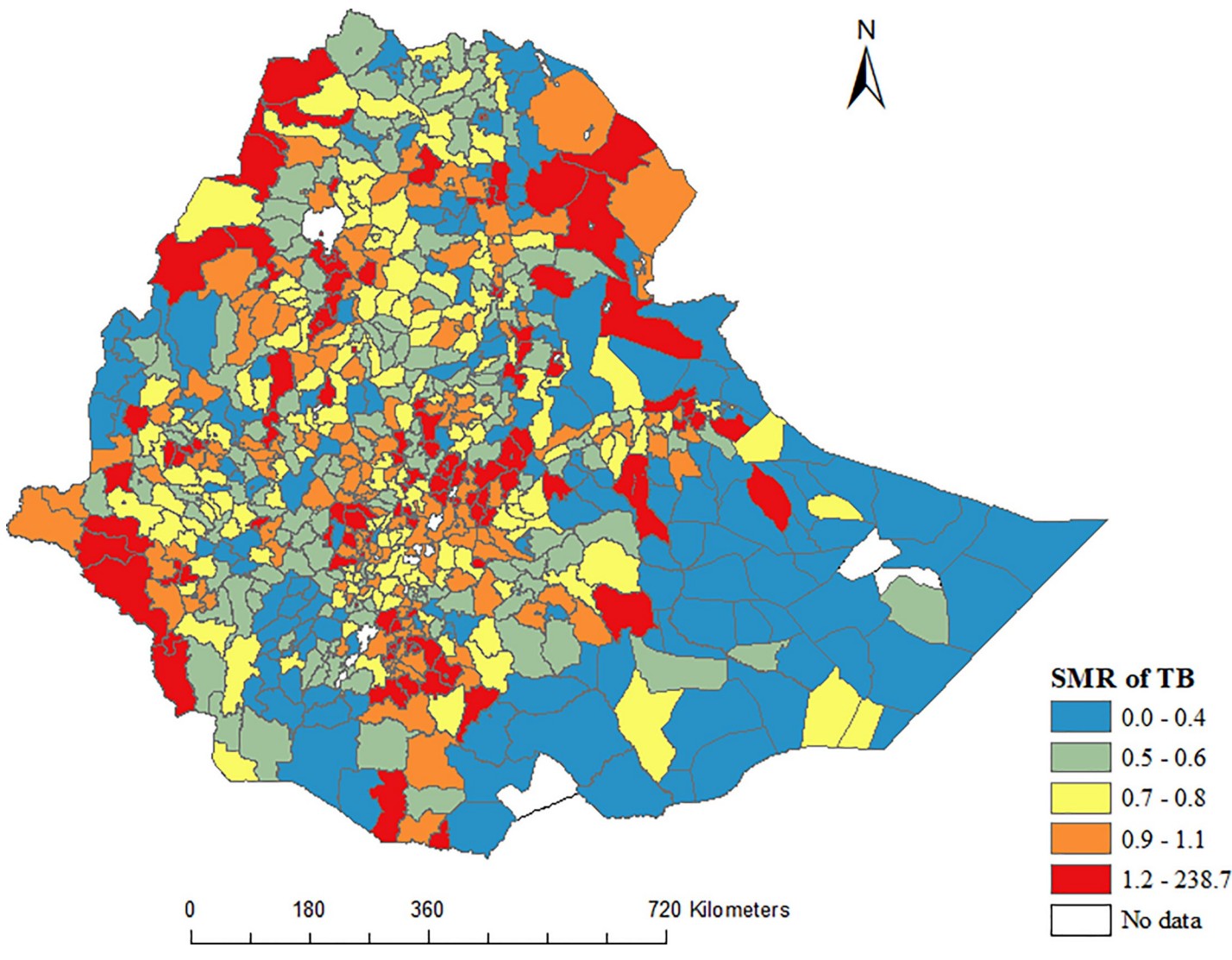

**Fig 2. Geographical distribution of the standardized morbidity ratios (SMR) of notified tuberculosis at district level in Ethiopia, 2016 to 2017.**

95%CI: 0.73, 0.96) (Table 2). The maps of the posterior means of the spatially structured random effects demonstrated evidence of spatial clustering of notified TB after accounting for the model covariates (Fig 4).

## Discussion

This is the first nationwide study to investigate the spatial distribution of notified TB in Ethiopia and to assess the role of health care access and behavioural factors on spatial clustering of notified TB. The result of this research showed that TB was spatially clustered in Ethiopia and there were significant relationships between incidence of notified TB and access to health care and knowledge about TB.

In this study, we found that the annual incidence rate of notified TB was 112 per 100,000 population, which is similar with the WHO TB 2018 report for Ethiopia (112.1 cases per 100,000 population) [1]. The incidence of TB reported in this study was higher compared to some other African countries, such as Egypt, Ghana, and Sudan [1, 2]. The high incidence rate

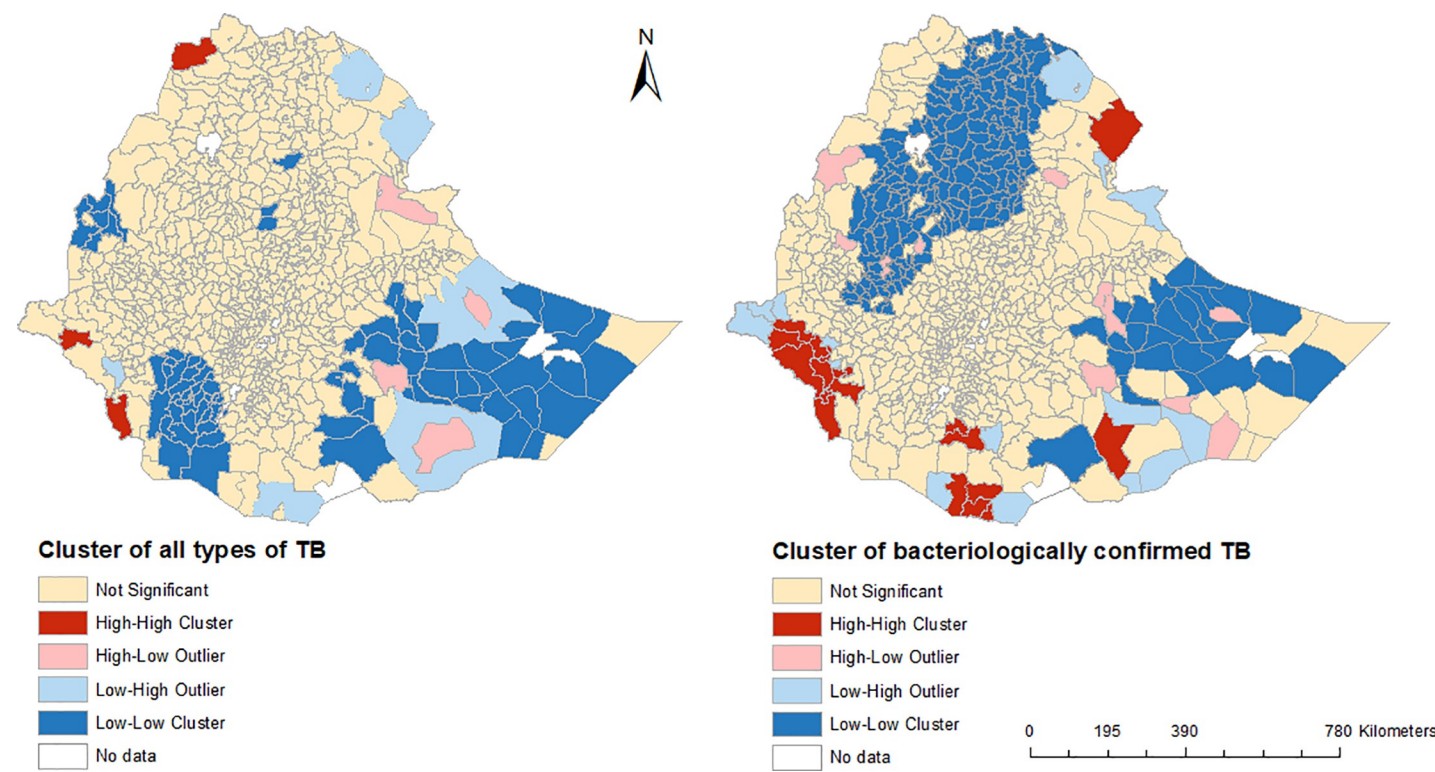

**Fig 3. Spatial clustering of bacteriologically confirmed and all forms of tuberculosis in Ethiopia based on the Local Moran's I statistic, between 2015 to 2017.**

of TB in Ethiopia presents a serious challenge for the national TB control efforts to achieve the global end-TB targets [10].

A higher incidence of notified TB was observed in city administration regions, including Addis Ababa (the capital city of Ethiopia), Dire Dewa (chartered city) and Harari (city administration region) and regional cities such as Adama, Jimma, and Bahirdar. This finding agrees with our previous studies in northwest Ethiopia where TB transmission and MDR-TB incidence rate were significantly associated with urbanization [6, 7]. Previous studies have also reported a relatively higher notification rate of TB in urban areas than in rural areas [11, 12]. There could be several reasons for this: the effect of urban residence on TB may be partially due to the higher HIV prevalence in urban areas [13–16]; and higher transmission could also occur in urban areas as a result of overcrowding and higher population density [15, 17]. On

**Table 2. Bayesian Poisson regression model with spatially structured and spatially unstructured random effects for notified incidence of tuberculosis in Ethiopia, 2016 to 2017.**

| Variables | Spatially unstructured model | Spatially structured model |
|---|---|---|
| | RR (95%CrI) | RR (95%CrI) |
| Low wealth index | 0.89 (0.74, 1.06) | 0.96(0.79, 1.152) |
| **Health care access problem** | **0.74 (0.63, 0.86)** | **0.78 (0.66, 0.90)** |
| Proximity to national borders | 1.17 (0.996,1.37) | 1.12(0.86, 1.46) |
| **Good knowledge about TB** | **0.73 (0.62, 0.84)** | **0.84 (0.73, 0.96)** |
| Constant (alpha) | -0.18(-0.32, -0.03) | -0.18(-0.19, -0.16) |
| Heterogeneity | 2.11 (1.51, 2.82) | 0.65 (0.47, 0.87) |
| DIC | 911.0 | 809.0 |

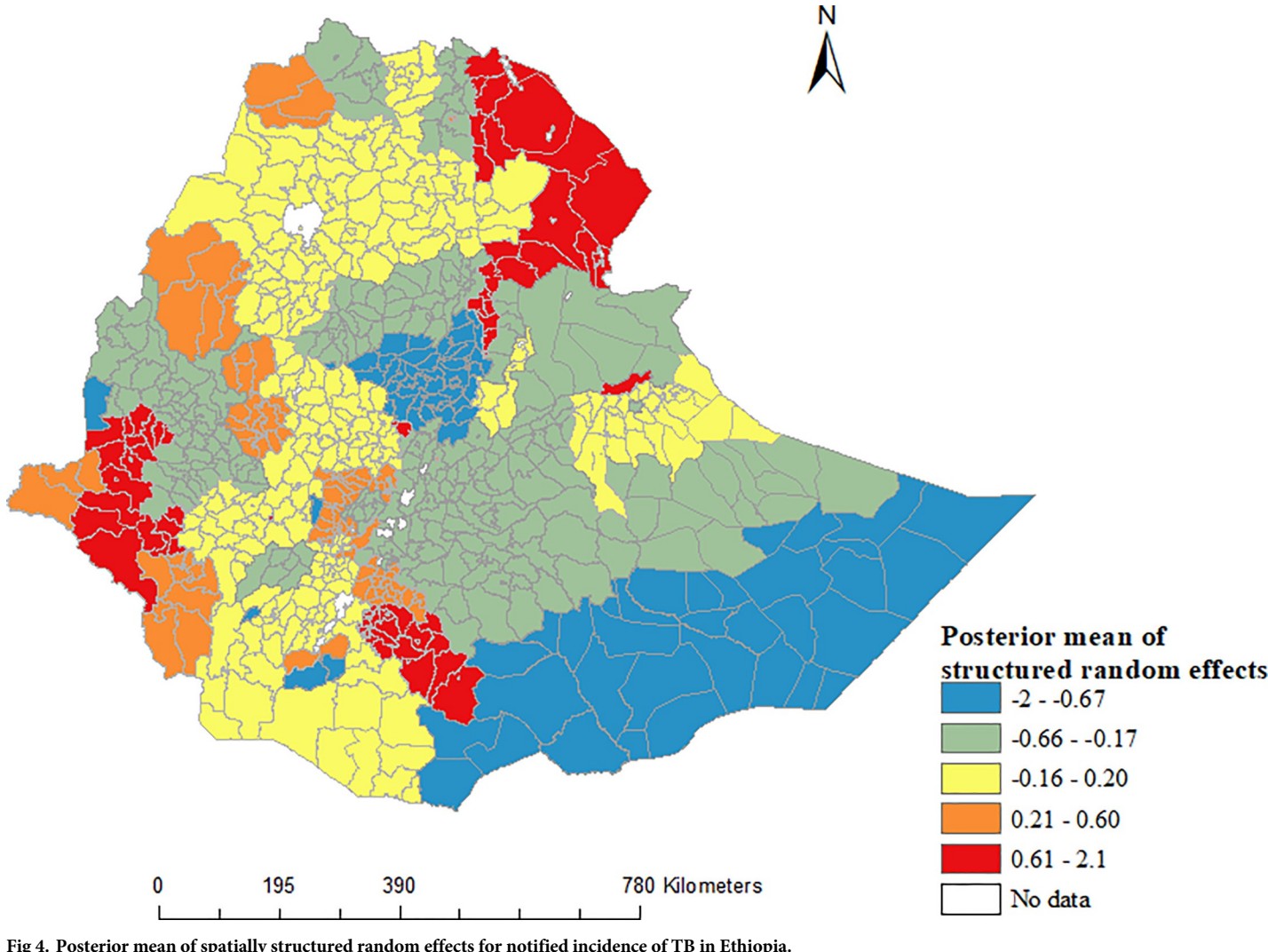

**Fig 4. Posterior mean of spatially structured random effects for notified incidence of TB in Ethiopia.**

the other hand, TB patients living in urban areas may have better access to TB diagnostic services and therefore more likely to report TB cases and be notified compared to patients living in rural areas where there is limited access to TB diagnostic services [18, 19].

High-high spatial clustering of notified TB was observed in districts located in the border regions and having high HIV prevalence [20]. Low-low clustering was observed in districts in Somali Region, a region with a lack of health care access and basic infrastructure [21].

Another finding of this study was that the incidence of notified TB was negatively associated with access to health care. This means that as the proportion of people in a district with difficulty of getting advice or treatment due to lack of money, or distance to a health facility increased, the TB notification rate decreased. This suggests that districts that had better access to health care were more likely to have better access to TB diagnostic and treatment services and therefore TB was more likely to be diagnosed and notified compared with districts that had poor access to health care [22]. Previous studies reported that a large proportion of TB patients remained undiagnosed because of a lack of access to health care services [23, 24].

Our study also showed that a unit increase in the percentage of people with good knowledge about TB was associated with a 16% decrease in TB risk. Creating awareness about the prevention and control of TB in the community has been the cornerstone of the National Tuberculosis Control Programme to reduce the incidence rate of TB [25–27]. However, according to the EDHS report, the knowledge of the community about the transmission and prevention of TB in Ethiopia was low; only 56% of women, 65% of men, 74% of urban residents and 62% of rural residents who had heard of TB knew that TB was spread through the air by coughing [28].

This study had some limitations. The accuracy of our estimates is primarily determined by the quality of HMIS data. Notably, the low TB case notification rates in some districts may not reflect the true burden of disease in these districts. A lack of health care services may be a barrier to TB patients being notified by the national TB surveillance system.

## Conclusion

This study showed substantial geographical variation in notified TB incidence in Ethiopia and provided maps that could be important source of evidence for planning TB control and prevention programs. High annual incidence rate of notified TB was observed in Ethiopia, and clustering of notified TB was found in some parts of the country. Ecological-level factors such as health care access and knowledge about TB were associated with the clustering of TB.

## Supporting information

**S1 Table. Summary of independent variables, sources of data and definition of variables.**
(DOCX)

**S2 Table. The incidence rate of bacteriologically confirmed and all forms of tuberculosis in Ethiopia at zone level and region level, between June 2016 and June 2017.**
(DOCX)

## Author Contributions

**Conceptualization:** Kefyalew Addis Alene, Archie C. A. Clements.

**Data curation:** Kefyalew Addis Alene.

**Formal analysis:** Kefyalew Addis Alene.

**Methodology:** Kefyalew Addis Alene, Archie C. A. Clements.

**Project administration:** Kefyalew Addis Alene.

**Supervision:** Archie C. A. Clements.

**Validation:** Kefyalew Addis Alene, Archie C. A. Clements.

**Visualization:** Kefyalew Addis Alene, Archie C. A. Clements.

**Writing – original draft:** Kefyalew Addis Alene.

**Writing – review & editing:** Kefyalew Addis Alene, Archie C. A. Clements.

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
