## [Decision Letter · Decision Letter 0]

25 Jul 2019

PONE-D-19-17592

Spatial clustering of notified tuberculosis in Ethiopia: a nationwide study

PLOS ONE

Dear Dr  Kefyalew Addis Alene

Thank you for submitting your manuscript to PLOS ONE. After careful consideration, we feel that it has merit but does not fully meet PLOS ONE’s publication criteria as it currently stands. Therefore, we invite you to submit a revised version of the manuscript that addresses the points raised during the review process.

We would appreciate receiving your revised manuscript by August 30th. To enhance the reproducibility of your results, we recommend that if applicable you deposit your laboratory protocols in protocols.io, where a protocol can be assigned its own identifier (DOI) such that it can be cited independently in the future. For instructions see: http://journals.plos.org/plosone/s/submission-guidelines#loc-laboratory-protocols

We look forward to receiving your revised manuscript.

Kind regards,

Massimo Ciccozzi

Academic Editor

PLOS ONE

Journal Requirements:

1. In ethics statement in the manuscript and in the online submission form, please provide additional information about the patient records/data used in your retrospective study. Specifically, please ensure that you have discussed whether all data were fully anonymized before you accessed them and/or whether the IRB or ethics committee waived the requirement for informed consent. If patients provided informed written consent to have their data used in research, please include this information.

2. We noticed you have some minor occurrence(s) of overlapping text with the following previous publication(s), which needs to be addressed:

https://doi.org/10.1186/s12879-019-4099-8

https://eprints.lancs.ac.uk/id/eprint/85146/1/2016DanielPhD.pdf.pdf

In your revision ensure you cite all your sources (including your own works), and quote or rephrase any duplicated text outside the Methods section. Further consideration is dependent on these concerns being addressed.

3. Please include a copy of Table 4 which you refer to in your text on page 7

Reviewers' comments:

Reviewer's Responses to Questions

**Comments to the Author**

1. Is the manuscript technically sound, and do the data support the conclusions?

Reviewer #1: Yes

Reviewer #2: Yes

2. Has the statistical analysis been performed appropriately and rigorously? 

Reviewer #1: Yes

Reviewer #2: Yes

3. Have the authors made all data underlying the findings in their manuscript fully available?

Reviewer #1: No

Reviewer #2: Yes

4. Is the manuscript presented in an intelligible fashion and written in standard English?

Reviewer #1: Yes

Reviewer #2: Yes

5. Review Comments to the Author

Reviewer #1: the article is technically sound and statistical analysis been performed appropriately and rigorously

Reviewer #2: Introduction: remove the word "end" in the phrase "short course (DOTS) strategy since 1994 and [5, 6]"

Discussion: change the word "visualisation" with "visualization"

6. PLOS authors have the option to publish the peer review history of their article (what does this mean?). If published, this will include your full peer review and any attached files.

Reviewer #1: No

Reviewer #2: Yes: Cecilia De Flora

---

## [Author Response · Author response to Decision Letter 0]

28 Jul 2019

Manuscript Number: PONE-D-19-17592

Article Title: Spatial clustering of notified tuberculosis in Ethiopia: a nationwide study

Authors: Kefyalew Addis Alene, Archie CA Clements

Journal: PLOS ONE

Response to reviewers' comments:

Reviewer #1: the article is technically sound and statistical analysis been performed appropriately and rigorously

Response: We greatly appreciate the reviewer’s efforts to carefully review the paper. 

Reviewer #2: Introduction: remove the word "end" in the phrase "short course (DOTS) strategy since 1994 and [5, 6]"

Response: This is now corrected.

Discussion: change the word "visualisation" with "visualization"

Response: This is also now corrected.

---

## [Editor Report · Decision Letter 1]

30 Jul 2019

Spatial clustering of notified tuberculosis in Ethiopia: a nationwide study

PONE-D-19-17592R1

Dear Dr. Kefyalew Addis Alene,

We are pleased to inform you that your manuscript has been judged scientifically suitable for publication and will be formally accepted for publication once it complies with all outstanding technical requirements.

With kind regards,

Massimo Ciccozzi

Academic Editor

PLOS ONE
---

## [Editor Report · Acceptance letter]

2 Aug 2019

PONE-D-19-17592R1 

Spatial clustering of notified tuberculosis in Ethiopia: a nationwide study 

Dear Dr. Alene:

I am pleased to inform you that your manuscript has been deemed suitable for publication in PLOS ONE. Congratulations! Your manuscript is now with our production department. 

With kind regards,

on behalf of

Prof Massimo Ciccozzi 

Academic Editor

PLOS ONE